# Identifying multicellular spatiotemporal organization of cells with SpaceFlow

Honglei Ren[1], Benjamin L. Walker [1,2], Zixuan Cang [3] & Qing Nie [1,2,4 ✉]

One major challenge in analyzing spatial transcriptomic datasets is to simultaneously incorporate the cell transcriptome similarity and their spatial locations. Here, we introduce SpaceFlow, which generates spatially-consistent low-dimensional embeddings by incorporating both expression similarity and spatial information using spatially regularized deep graph networks. Based on the embedding, we introduce a pseudo-Spatiotemporal Map that integrates the pseudotime concept with spatial locations of the cells to unravel spatiotemporal patterns of cells. By comparing with multiple existing methods on several spatial transcriptomic datasets at both spot and single-cell resolutions, SpaceFlow is shown to produce a robust domain segmentation and identify biologically meaningful spatiotemporal patterns. Applications of SpaceFlow reveal evolving lineage in heart developmental data and tumor-immune interactions in human breast cancer data. Our study provides a flexible deep learning framework to incorporate spatiotemporal information in analyzing spatial transcriptomic data.

[1] The NSF-Simons Center for Multiscale Cell Fate Research, University of California Irvine, Irvine, CA 92627, USA. [2] Department of Mathematics, University of California Irvine, Irvine, CA 92627, USA. [3] Department of Mathematics, North Carolina State University, Raleigh, NC 27695, USA. [4] Department of Developmental and Cell Biology, University of California Irvine, Irvine, CA 92627, USA. ✉email: qnie@uci.edu

The spatiotemporal pattern of gene expression is critical to unraveling key biological mechanisms from embryonic development to disease. Recent advances in spatially resolved transcriptomics (ST) technologies provide new ways to characterize the gene expression with spatial information that the popular nonspatial single-cell RNA-sequencing (scRNA-seq) method is unable to capture. The majority of current ST technologies may be categorized into in situ hybridization (ISH)-based and spatial barcoding-based, varying in gene throughput and resolution[1–3]. ISH-based methods can detect target transcripts at the sub-cellular resolution, such as Multiplexed Error-Robust Fluorescence ISH (MERFISH) and sequential fluorescence ISH (seqFISH), for about 100–1000 and 10,000 genes respectively[4,5]. Spatial barcoding-based methods can capture the whole transcriptome with varying spatial spot resolutions, such as Visium in 55 μm, the Slide-seq in 10 μm[6], and the spatiotemporal enhanced resolution 'omics sequencing (Stereo-seq) in nanometer (subcellular) resolution[7].

Compared to non-spatial technologies such as scRNA-seq, the presence of spatial information in ST data necessitates development of methods that natively handle the high-dimensional features in space. High-dimensional spatially aware analyses have been previously explored largely in the context of image data[8–11]. By considering each gene as one channel in an image, spatial transcriptomic data may be abstracted as high-dimensional images. However, uncovering the biological interactions between genes in tissue requires new computational methods tailored specifically to transcriptomic data.

Many methods developed for non-spatial transcriptomic data such as scRNA-seq or bulk spatial transcriptomics data[12,13] may provide insights in designing approaches for ST data at single-cell resolution through recasting the relevant tasks in a spatial manner. For example, the identification of spatially variable genes in ST data[14,15] can be viewed as the spatial extension of the highly variable genes in scRNA-seq data. Similarly, methods have been developed to identify spatial domains in ST data[16], the analog of cell clustering in scRNA-seq data analysis, but using spatial information to produce spatially coherent regions. Giotto[17], BayesSpace[18], and SC-MEB[19] use Markov random fields to model the related gene expression in neighboring cells. stLearn utilizes morphological information to perform spatial smoothing before clustering[20]. MULTILAYER uses graph partitioning to segment tissue domains[21]. MERINGUE performs graph-based clustering using a weighted graph that combines spatial and transcriptional similarity[22]. SpaGCN[23], SEDR[24], SCAN-IT[25], stMVC[26], and STAGATE[27] build deep auto-encoder networks to learn low-dimensional embeddings of both gene expression and spatial information, and segment domains through embedding clustering. RESEPT learns a three-dimensional embedding from ST data by a spatial retained graph autoencoder and treats the embedding as a 3D image, identifying domains through image segmentation using a convolutional neural network[28].

The domain segmentation methods reviewed above are the ST counterpart of cell clustering in scRNA-seq data analysis. Contrary to discrete clustering, another powerful analysis in scRNA-seq is the concept of continuous pseudotime which can represent developmental trajectories. The dynamics of many developing systems such as regeneration and cancer progression are often spatially organized[29,30]. The ST data thus provides an opportunity to simultaneously reveal both spatial and temporal structures of development. While pseudotime methods for scRNA-seq can be directly applied to ST data, the resulting trajectory may be discontinuous in space. stLearn combines nonspatial pseudotime with spatial distance by simple average, as well as filters connections between clusters inferred by scRNA-seq trajectory inference methods using a spatial distance cutoff, but the resulting connections are limited by the initial pseudotime trajectories inferred without using spatial information[20]. There is thus a demand for computational tools for integrative reconstruction of fine-resolution spatiotemporal trajectories from ST data which is continuous both in time and space.

As pseudotime trajectories are traditionally computed from a low-dimensional embedding of transcriptomic data[31], the computation of spatiotemporal trajectories can be viewed as a problem of constructing spatially aware embeddings of ST data. Multiple strategies for computing spatially aware embeddings may be used such as Hierarchical SNE[32], Hierarchical UMAP[33], dual embedding[34]. Additionally, deep graph neural network-based approaches, such as DeepWalk[35], Variational Graph Auto-Encoder (VGAE)[36], Graph2Gauss[37], and Deep Graph Infomax (DGI)[38], while computationally more expensive, have been utilized for ST data due to their flexibility to model and learn nonlinear and complex salient spatial dependencies between genes and cells.

In this work, we develop a framework to reveal continuous temporal relationships with spatial context using ST data. By combining a DGI framework with spatial regularization designed to capture both local and global structural patterns, we extract a spatially consistent low-dimensional embedding and construct a pseudo-Spatiotemporal Map (pSM), representing a spatially coherent pseudotime ordering of cells that encodes biological relationships between cells, along with a region segmentation. We compare SpaceFlow with five existing methods on six ST datasets, demonstrating competitive performance on benchmarks, and use SpaceFlow to reveal evolving cell lineage structures, spatiotemporal patterns, cell-cell communications, tumor-immune interfaces and spatial dynamics of cancer progression.

## Results

**Overview of SpaceFlow**. SpaceFlow takes Spatial Transcriptomic (ST) data as input (Fig. 1a) and outputs a spatially consistent low-dimensional embedding, domain segmentation, and pseudo-Spatiotemporal Map (pSM) of the tissue. The input ST data consists of an expression count matrix and spatial coordinates of cells or spots. The output embedding encodes the expression of ST data so that nearby embeddings in the latent space reflect not only the similarity in expression but also spatial proximity. The domain segmentation characterizes the spatial patterns of tissue without the need for histological or pathological knowledge. The pSM is a map that represents the pseudo-spatiotemporal relationship of cells in ST data.

Before applying the deep graph network, a Spatial Expression Graph (SEG) is constructed (Fig. 1b) with nodes in the graph representing cells with expression profile attached, while edges model the spatial adjacency relationship of cells (Fig. 1b). In addition, an Expression Permuted Graph (EPG) is constructed by randomly permuting the nodes in SEG and used as negative inputs for the network. To encode the SEG into low-dimensional embeddings, a graph convolutional encoder is built with Parametric ReLU (PReLU) as activation (Fig. 1c). The graph convolutional encoder applies a weighted aggregation to the expression of a cell with its spatial neighborhood to capture local expression patterns into embeddings. We utilize a Deep Graph Infomax (DGI) framework to train the encoder[38], which optimizes a Discriminator Loss (Fig. 1d bottom) to learn to distinguish the embeddings from SEG and EPG input. Compared to other GCN architectures, this allows the encoder to learn embeddings that emphasize specifically the spatial expression patterns that corresponding to meaningful structure as opposed to those due to non-spatial variation or noise.

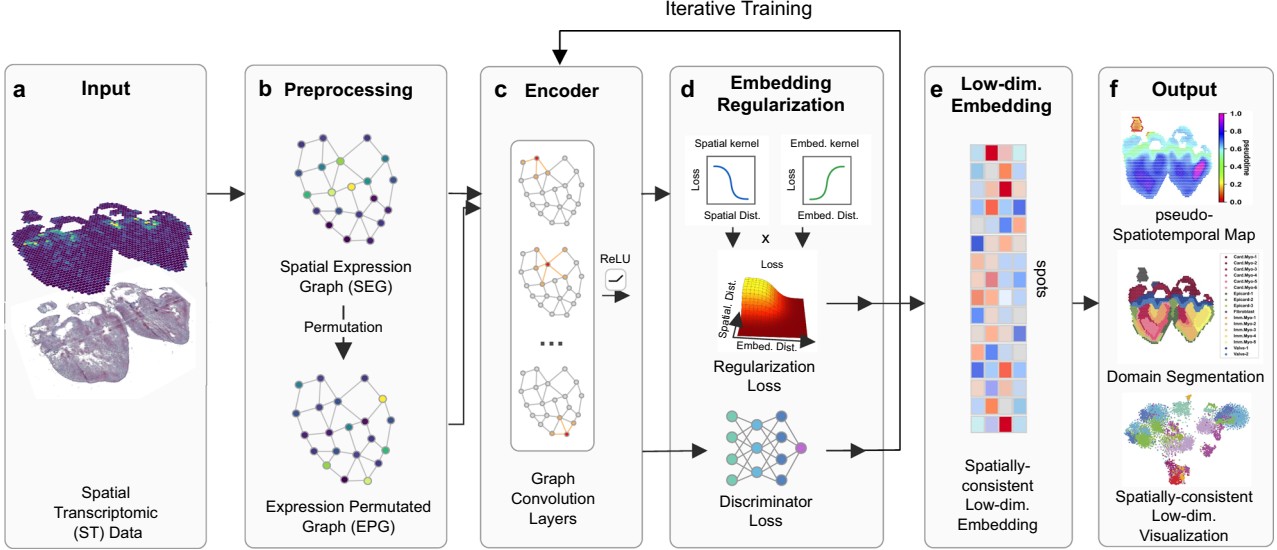

**Fig. 1 Overview of SpaceFlow. a** The input ST dataset consist of an expression count matrix and spatial coordinates of spots/cells. **b** A spatial expression graph (SEG) is constructed as the network input, with edges characterizing the spatial neighborhood, and nodes representing cells/spots with expression profiles attached. By randomly permuting the nodes in SEG, Expression Permuted Graphs (EPG) are built as negative samples. **c** A two-layer GCN encodes the SEG or EPG input into low-dimensional embeddings. **d** The embeddings are regularized for spatial consistency. With the Spatial Regularization loss and the Discriminator loss, the encoder is iteratively trained until convergence. **e** The low-dimensional embedding is obtained from the trained encoder. **f** The output consists of the pseudo-Spatiotemporal Map (pSM), domain segmentation, and the visualization of low-dimensional embeddings.

Distant cells of the same cell type may exhibit a high degree of transcriptional similarity even when in very different parts of tissue. Consequently, in order to produce embeddings that most meaningfully represent spatial structure, one needs spatial consistency in embeddings, meaning that the latent space embeddings should be distant not only if their expression profile is distinct, but also when the expression is similar but their spatial locations are distant. We use embedding regularization to enforce this structure in the latent space (Fig. 1d), which takes the spatial distance matrix and the embedding distance matrix of cells or spots as input. These two matrices are then input into linear kernels (Spatial kernel and Embedding kernel) to calculate the loss of each cell pair based on the spatial or embedding distance. The spatial losses and embedding losses of cells from these kernels are then combined to produce the final regularization loss which is added to the discriminator loss used to train the encoder (Methods). The learned low-dimensional embeddings (Fig. 1e) for the ST data are then used in downstream analysis, including the pseudo-Spatiotemporal Map (pSM), domain segmentation, and low-dimensional visualization (Fig. 1f) to analyze spatio-temporal patterns of tissues.

**Comparison of SpaceFlow with five existing methods for ST data at spot resolution**. To evaluate the quality of the SpaceFlow embeddings, we compared it with five existing methods for unsupervised segmentation on ST data: one non-spatial method Seurat v4[39], and four spatial methods Giotto[17], stLearn[20], MERINGUE[22], and BayesSpace[18] on a 10x Visium human Dorso-Lateral Pre-Frontal Cortex (DLPFC) dataset consisting of twelve samples[40]. Spots are annotated as one of six layers (layer 1 through layer 6) or white matter, and these annotations are used as the ground truth for benchmarking.

To compare the domain segmentation performance quantitatively, we used the adjusted Rand Index (ARI) to measure the similarity between the inferred domains and the expert annotations across all twelve sections (Fig. 2a). SpaceFlow shows a 0.427 median ARI score, the second-highest across the six methods,

slightly lower than the BayesSpace, which has 0.438 median ARI. MERINGUE shows the lowest median ARI score (0.232), followed by Seurat (0.300) and then Giotto (0.332), and stLearn (0.369). Interestingly, the DGI method without the spatial regularization used in SpaceFlow shows a significant decrease in ARI, with a 0.332 median score, indicating that spatial regularization does effectively improve the domain segmentation of DGI.

Next, we performed a more detailed analysis on section 151671 (Fig. 2b–f). We first computed the domain segmentation for each method and visualized the output compared to the expert annotation (Fig. 2b). It is seen that all methods fail to capture the subtle structure of Layer 4 (L4), suggesting that this ST data does not have the necessary spatial resolution to capture the L4 structure. Both SpaceFlow and BayesSpace can capture all the remaining structures (L3, L5, L6, and WM) observed in the annotation. Moreover, BayesSpace identified the outer ring of the WM as an additional structure, whereas SpaceFlow found a different structure at the top right part of Layer 3 (labeled as domain 3 in orange). The structure found in SpaceFlow is consistent with the domains from Giotto, stLearn, and MER-INGUE. stLearn also identified L5, L6, and WM that are consistent with the annotation but with noisy boundaries between domains. Giotto and MERINGUE identified the L6 and WM domain but are unable to identify the boundary of L5 along with the same noisy boundary issues. Seurat showed an overall disordered domain structure and can barely capture the white matter (WM) structures. The DGI method (Fig. 2b), like SpaceFlow but without spatial regularization, showed a layered structure with inconsistent boundary shape and non-contiguous domains, reinforcing the importance of spatial regularization. Similar results can also be observed in samples 151507 and 151673 (Supplementary Fig. 1a, c).

We next compared the low-dimensional embeddings from Seurat, stLearn, DGI, and SpaceFlow (the segmentation methods Giotto, MERINGUE, and BayesSpace do not produce embeddings), applying UMAP to the embeddings to produce a two-dimensional visualization of all spots colored by the layer

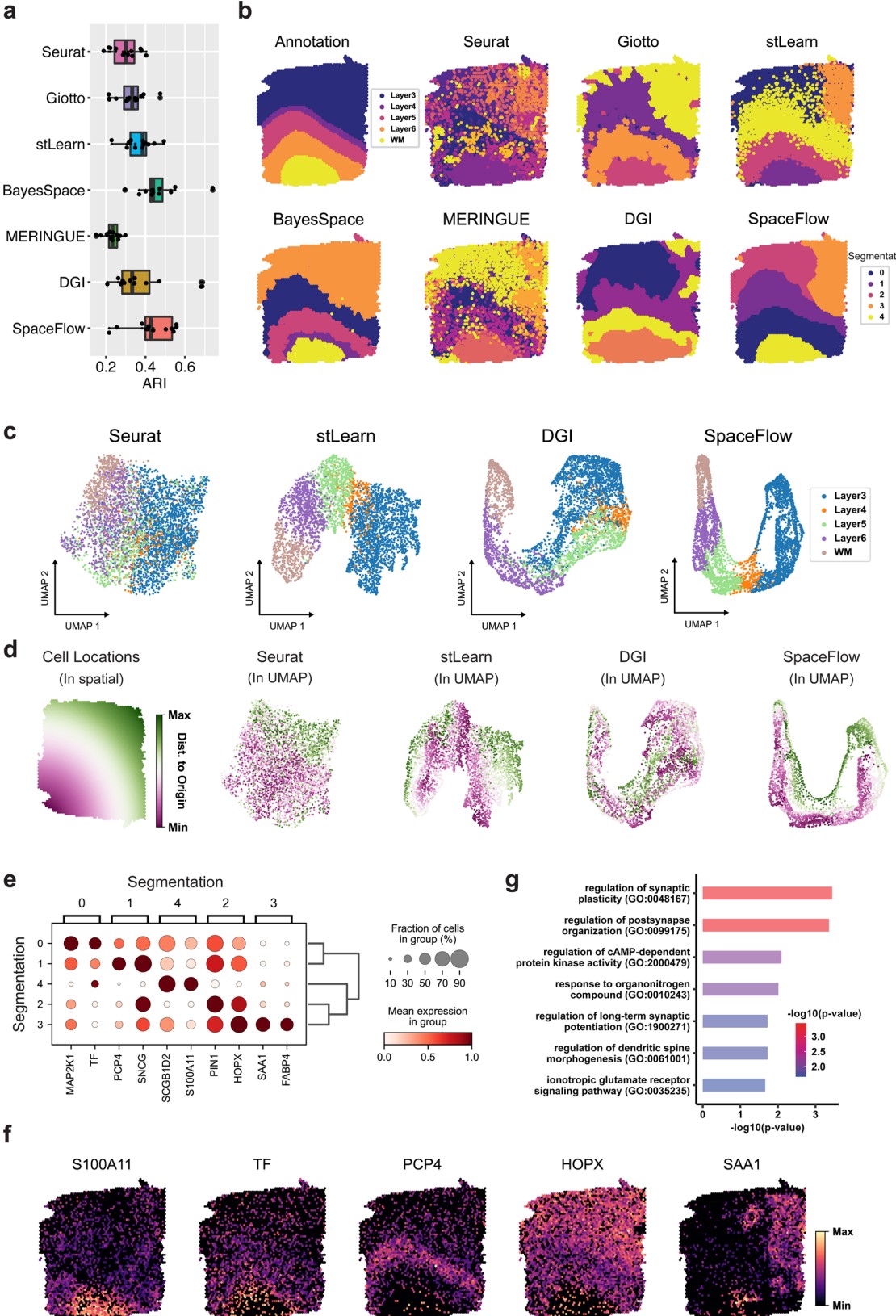

annotation. We observe that SpaceFlow embeddings produce embeddings that clearly separate the spots by layer when compared to stLearn, DGI, and Seurat (Fig. 2c). As the separation between low-dimensional embeddings of the regions provides an upper limit on the ability of segmentation to separate the regions, this shows that the incorporation of spatial regularization

produces more distinct embeddings between different layers and thereby a greater ability to distinguish them in downstream analysis.

To study how the low-dimensional embeddings from different methods encode spatial information, we show the same UMAP embeddings colored by the spatial distances between the spot and

**Fig. 2 Comparison with five unsupervised methods shows that SpaceFlow can identify biologically meaningful spatial domains and generate spatially consistent low-dimensional embeddings. a** Boxplot of clustering accuracy in all sections of the LIBD human dorsolateral prefrontal cortex (DLPFC) ST dataset[41] ($n = 12$ sections) in terms of adjusted rand index (ARI) scores for seven methods. In the boxplot, the center line, box limits and whiskers denote the median, upper and lower quartiles, and 1.5× interquartile range, respectively. **b** Domain segmentations of cortical layers and white matter by annotation (top left panel) and by seven different methods (other panels) using section 151671 of DLPFC data. **c** UMAP visualizations for DLPFC data section 151671, using low-dimensional embeddings from Seurat, stLearn, DGI, and SpaceFlow colored by the layer annotation of spots. **d** Cell spatial locations (left panel) and UMAP visualizations (right four panels) colored by the Euclidean distance between spot/cell and origin (0,0), which is the left bottom corner of the first panel. **e** Dot plot of the gene expression of domain-specific markers. The dot size represents the fraction of cells in a domain expressing the marker and the color intensity represents the average expression of the marker in that domain. **f** Spatial expression for the top-1 markers of the identified domains. **g** The Gene Ontology (GO) enrichment analysis of the domain-specific genes (161 genes) for the domain 3 in panel **b**, SpaceFlow. Both the color and the length of bars represent the enrichment of GO terms using -log10(p-value) metric from topGO analysis. *P* values were obtained using the one-sided Fisher's exact test without multiple-testing correction. *P* values < 0.001 were considered significant.

the origin (Fig. 2d), so that embeddings which preserve the spatial structure will maintain this color gradient. Seurat embeddings exhibit a high level of mixing in this color, indicating significant deformities in both local and global structure, whereas DGI shows local color gradients with a minor color mixture and shows no global gradient structure. In contrast, SpaceFlow and stLearn both exhibit a clear global gradient structure with a clear coloring trend in each annotation layer. This indicates the learned embeddings from SpaceFlow and stLearn encode transcriptional information while also preserving the local and global spatial structure of the data.

To check whether the identified domains from SpaceFlow are biologically meaningful, we performed a domain-specific expression analysis. We found spatial specific expression patterns for the identified domain-specific genes. For instance, the top domain-specific gene for domain 3 (orange) in SpaceFlow is SAA1. Within domain 3, it is expressed in 90% of cells with a mean expression of 0.96 (scaled from 0 to 1), whereas outside this domain it is expressed in less than 30% of cells with a mean expression of approximately 0.13 (Fig. 2e). This spatial expression specificity is also clear in the spatial expression heatmap showing top-1 marker genes for each domain (Fig. 2f). Among these genes, we found that PCP4 was previously reported as the marker for layer 6 in prefrontal cortex[41]. The other genes have clear layer correlations although not previously reported, suggesting new experiments are needed for validating the potential new marker genes. We carried out a Gene Ontology (GO) analysis for the domain-specific genes whose *p* value is less than 0.01 in domain 3 (Fig. 2g). We observed GO terms associated with regulation of cAMP-dependent protein kinase activity, ionotropic glutamate receptor signaling pathway, regulation of long-term synaptic potentiation regulation of synaptic plasticity, etc. This suggests that the spatially specific expression in the identified domain 3 may be related to long-term synaptic activity, which is consistent with the observation that several top domain-3 specific expression patterns such as MALAT1 (*p* value < 5.5e−33)[42], CAMK2A (*p* value < 7.4e−23)[43], PPP3CA (*p* value < 4e−16)[44] are involved in long-term synaptic potentiation. The fact that this gene expression is clearly related to neural activity indicates a meaningful subdivision of Layer 3 despite a lack of annotations for this region. This suggests that the expert annotations, even if accurately describing the layer structure, may not paint a complete picture of the spatial structure within the data.

**SpaceFlow uncovers pseudo-spatiotemporal relationships among cells.** Next, we study the pseudo-Spatiotemporal Map (pSM) computed by SpaceFlow. Different from traditional pseudotime as used in scRNA-seq analysis, which only considers the similarity in expression between cells, the pSM considers both spatial and transcriptional relationships among cells simultaneously (Methods). In spatial visualizations of the pseudotimes

produced from Seurat, Monocle, traditional single-cell pseudo-time methods that do not incorporate spatial information, we observed a lack of layered patterns as well as significant noise (Fig. 3a). In contrast, both spatially aware methods tested, stLearn and SpaceFlow present a layer-patterned pSM with a clear and smooth color gradient (Fig. 3a), suggesting a pseudo-spatiotemporal ordering from White Matter (WM) to Layer 3. This ordering mirrors the correct inside-out developmental sequence of cortical layers and reflects the layered spatial organization of the tissue. However, stLearn shows less consistency with the annotation in the White Matter (WM) region when compared to SpaceFlow. Similar patterns can also be observed in samples 151507 and 151673 (Supplementary Fig. 1b, d). We also run SpaceFlow on more 10x Visium ST datasets, the results can be found in Supplementary Information (Supplementary Fig. 3).

To test the capability of SpaceFlow on single-cell resolution ST data with a large number of cells, we evaluated SpaceFlow on a Stereo-seq dataset from mouse olfactory bulb tissue, capturing 28243 genes across 18197 cells[7], comparing with traditional pseudotime methods Seurat, Monocle, and Slingshot. We observed that Seurat shows little variation in pseudotime across the tissue except for the outer rings, which show slightly higher pseudotime values than in the inside. Monocle is much noisier than Seurat and shows no clear patterns. Slingshot is similar to Seurat and exhibits outer-ring patterns. By contrast, SpaceFlow presents a clear layered pattern mirroring the annotated layers of the olfactory bulb tissue (Fig. 3b). The pSM value (red) is lowest in the external plexiform layer (EPL) and then increases when moving away in both directions. This ordering in the pSM is consistent with the developmental sequence of these layers, where starting from the central EPL, development proceeds bilaterally outwards, leading to the mitral cell layer (MCL) and glomerular layer (GL), olfactory nerve layer (ONL), and the granule cell layer (GCL) develops last[45]. The highest values are observed on the inner side, with the peak in the granule cell layer (GCL) and the rostral migratory stream (RMS). This shows that the pSM computed by SpaceFlow is not only more clearly spatially organized than non-spatial pseudotime, but that these results also more accurately represent the temporal and developmental relationships between cells.

We next identify marker genes from the pSM. By calculating the top genes by correlation with the pSM values, we found genes that are predominantly expressed in layers of the olfactory bulb tissue (Fig. 3c). One of the top marker genes, NRGN, shows clear expression patterns localized in the granule cell layer (GCL), and previous experiments have shown that NRGN is usually expressed in granule-like structures in pyramidal cells of the hippocampus and cortex[46]. This shows how the pSM can be used to facilitate biomarker identification for tissues.

Next, we compared the domain segmentation performance of SpaceFlow against Seurat, running without incorporating spatial

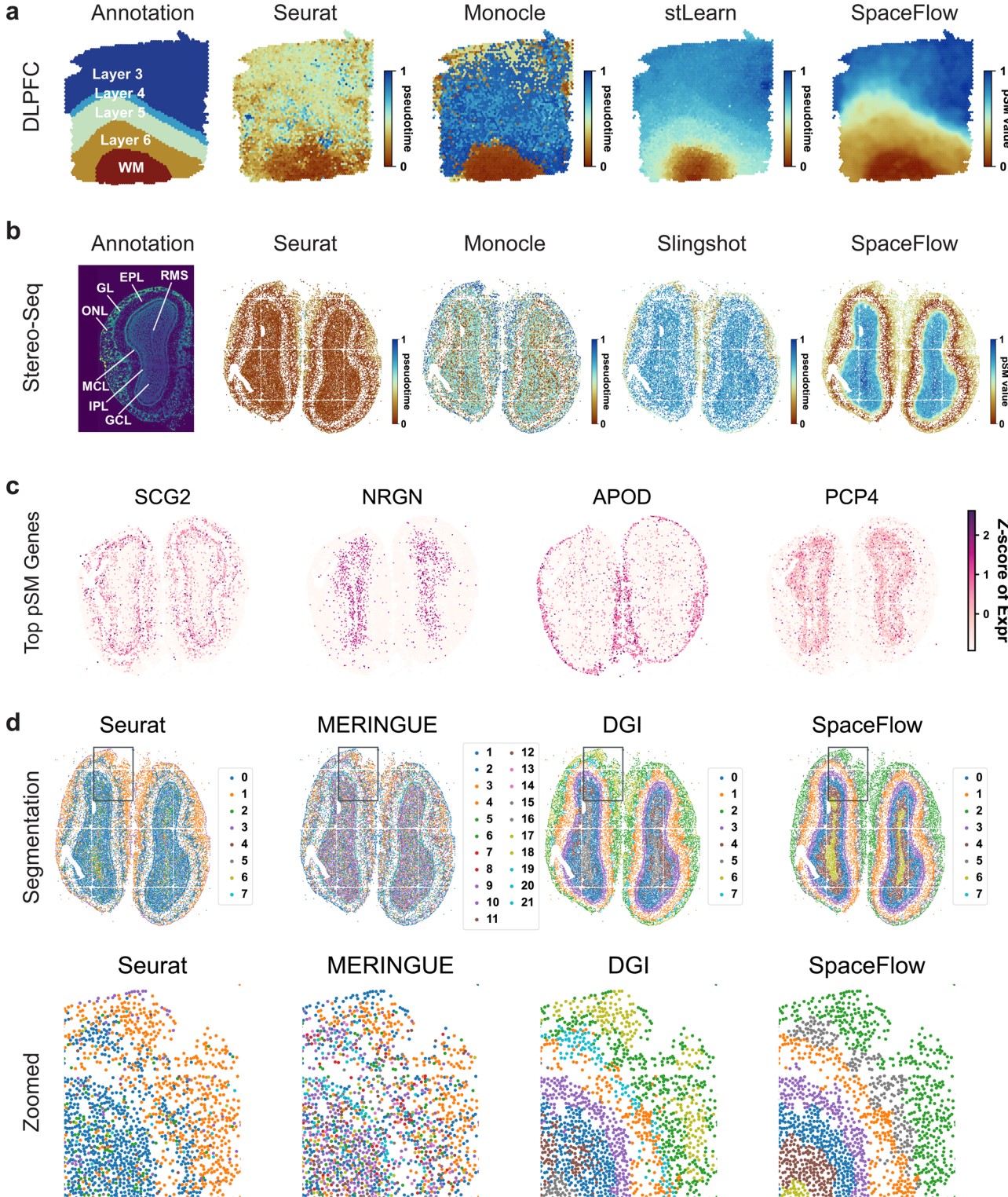

**Fig. 3 SpaceFlow generates pseudo-Spatiotemporal Map for ST data and uncovers pseudo-spatiotemporal relationship between cells in both spots-resolution and single-cell resolution ST data. a** Spatial visualization of pseudotime calculated by Seurat, Monocle, stLearn, and the pSM generated by SpaceFlow on DLPFC data (same dataset as in Fig. 2, spot resolution). **b** Spatial visualization of pseudotime calculated by Seurat, Monocle, Slingshot, and the pSM generated by SpaceFlow on Stereo-seq data (single-cell resolution). **c** Spatial expression of genes exhibiting the highest correlation between expression and the pSM value of corresponding spots/cells. The color represents the z-score of expression level. **d** Domain segmentations of Stereo-seq ST data given by Seurat, MERINGUE, DGI, and SpaceFlow. Top row: full views of the domain segmentations from different methods, bottom row: zoomed views of regions boxed in each panel.

data, as well as the spatial methods MERINGUE and DGI on the Stereo-seq data. We show a global and a zoomed view of the identified domains for each method (Fig. 3d). The Seurat segmentation is characterized by two large regions – all inner layers except the olfactory nerve layer (ONL) are mainly combined into one region (domain 0 in dark blue) and the ONL is segmented as another (domain 1 in orange). However, even in the inner layers, there are many cells classified as domain 1 (orange), lacking a clear separation between domains. To control for the effect of the resolution parameter, we considered values of 0.3 to 0.8, 1.0 and 2.0, resulting in 13, 15 and 30 clusters respectively (Supplementary Fig. 2a). However, the spatial consistency of clusters does not improve with a higher resolution parameter. MERINGUE identified three major layers, with one additional compared to Seurat, which corresponds to the external plexiform layer (EPL) in the annotation in Fig. 3b; however, there is significant spatial noise and it is difficult to see boundaries between tissue layers even in the zoomed view. With the DGI method, we observed a layered structure of domains, but significant mixing of domains is still visible in the zoomed view. In SpaceFlow, the eight-layer structure is much clearer, as nearly no mixture between occurs the corresponding ONL (domain 2 in green) and GL (domain 5 in silver gray) regions, whereas there are clear mixtures of labels by the other methods across all regions.

**SpaceFlow reveals evolving cell lineage structures in chicken heart development ST data.** To study how the pSM may be used to uncover spatial expression dynamics in embryonic development, we retrieved and utilized an ST dataset on the chicken heart[29] at four key Hamburger-Hamilton ventricular developmental stages. The dataset contains 12 tissue sections in total, and is sequenced at day 4 (5 sections), day 7 (4 sections), day 10 (2 sections) and day 14 (1 section). To build a baseline for comparison, we first visualized spot annotation from the original study (Fig. 4a). Then, we computed the domain segmentation from SpaceFlow for each time point (Fig. 4b) and labeled the domains based on their top marker genes as compared with the literature (Details in Supplementary Data 1).

We first found an evolving lineage structure, annotated as Valve in Fig. 4b. This newly identified structure is evident from Day 7 (D7) with a layered structure and consistent shape during heart development. The identified structure is consistent with the anatomical regions of the chicken heart at the sequenced stages[29,47]. We also characterized the transition dynamics of the myocardium from the immature to the mature state across the period from D4 to D14 (immature myocardium annotated by orange/yellow change into cardiomyocytes annotated in red/pink). Moreover, we identified that the epicardium structure (annotated in green) on the outer ring of immature myocardium transformed into cardiomyocytes from D7 to D14.

To better understand the spatiotemporal organization of the chicken heart during development, we computed the pSM for each time point separately (Fig. 4c), considering that pseudo-time across tissues with different time points may not be comparable. Similar to the domain segmentation, the identified valve structures are clear from D7 to D14 in the pSM. In addition, the myocardium in the ventricles is more homogeneous in the pSM (blue) than in the domain segmentation. This suggests the difference in the myocardium of ventricles might be much more subtle than regions showing different pSM values. We also found the annotated myocardium in ventricles to consistently show higher pSM values (blue) than other regions, which indicates the pseudo-spatiotemporal ordering of the myocardium in the ventricles is later than other regions in the same stage. By

contrast, the identified valve structures show yellow color in the pSM from D7 to D14, suggesting the ordering is relatively late compared with the regions colored in red or orange. By plotting pSM values of spots against the first component of the UMAP embedding (Fig. 4d), similar patterns can be observed, where the cells with valve annotations colored in blue shows intermediate pSM values (y-axis) and lies in the middle of the trajectories in Fig. 4d. These spatiotemporal patterns revealed in the pSM are consistent with previous observations in chicken cardiac development[47].

Through a hierarchical clustering for domains across all four stages based on the expression of top domain-specific marker genes, we found expression programs specific to evolving structures (Fig. 4e). We observed the valves of D7 and D10 to be similar to each other in expression, with genes that regulate cell growth and proliferation such as S100A11, S100A6, and CNMD, as well as genes associated with cell-collagen interaction such as TGFBI, found as the top marker genes for these populations. We also performed GO analysis to study the function of identified valve structures (Fig. 4f) and found enrichment of GO terms associated with negative regulation of BMP signaling pathway and negative regulation of epithelial-mesenchymal transformation (EMT). Previous studies found that EMT mediated by BMP2 is required for signaling from the myocardium to the underlying endothelium to form endocardial cushion (EC), which ultimately gives rise to the mature heart valves and septa[48]. We also observed enrichment of positive regulation of canonical Wnt signaling pathway, previously shown as a regulator of endocardial cushion maturation as well as valve leaflet stratification, homeostasis, and pathogenesis[49].

To investigate cell-cell communication between the identified valve structures and other tissue regions, we performed space-constrained CellChat analysis[50] using the domain labels from SpaceFlow as groupings. The top two identified pathways for the valve structures are midkine (MK) and pleiotrophin (PTN), which belong to the subfamily of heparin-binding growth factors. We observed strong signaling in MDK-SDC2, MDK-NCL, PTN-SDC2, and PTN-NCL ligand-receptor pairs from valve tissue to nearby immature cardiomyocytes and atrium cardiomyocytes (Fig. 4g). These interactions have various functions, such as angiogenesis, oncogenesis, stem cell self-renewal, and play important roles in the regeneration of tissues, such as the myocardium, cartilage, neuron, muscle, and bone[51]. Studies have shown that midkine impedes the calcification of aortic valve interstitial cells through cell-cell communications[52]. In addition, SDC2 is found required for migration of the bilateral heart fields towards the mid-line in zebrafish model[53]. Pleiotrophin (PTN) is usually considered a cytokine and growth factor that promotes angiogenesis[54]. Together, the observed cell-cell communication based on the structures identified by SpaceFlow suggests anti-calcification and pro-angiogenesis processes are important during the maturation of valve tissue.

**SpaceFlow identifies tumor-immune microenvironment in human breast cancer ST data.** To study the cancer microenvironment interaction and tumor progression, we applied SpaceFlow to human breast cancer ST data[55]. We show here results for sample G, consistent with the original paper. Results for other samples can be found in the Supplementary Information (Supplementary Fig. 6). First, we performed domain segmentation and compared it to the expert annotation (Fig. 5b). The obtained domains were labeled based on their marker genes (Details in Supplementary Data 2). The regions in-situ cancer-1/2, APC, B,T-1/2, and invasive-1/2 identified in the SpaceFlow segmentation agreed with the annotations Immune rich,

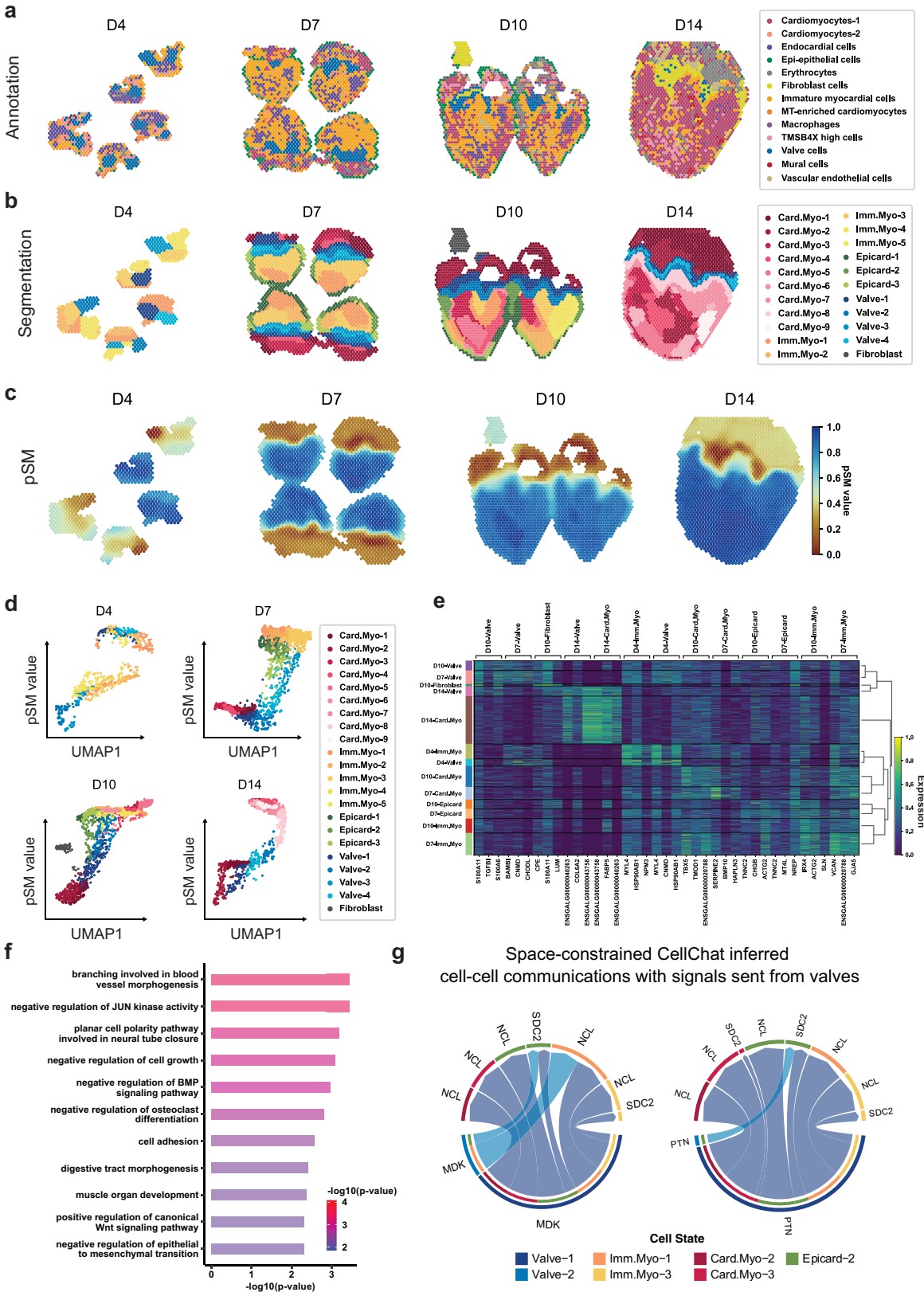

Immune:B/plasma, and Cancer 1 respectively from the original study. However, we also identified three tumor-immune interface regions, labeled as Tu.Imm.Itfc-1/2/3, which were labeled as mixtures of other cell types in the original study.

To reveal the pseudo-spatiotemporal relationship between spots in tissues, we generated the pSM and compared it with the spatially visualized pseudotime calculated by two alternatives: Monocle, which does not use spatial information, and applying DPT to spatially aware embeddings from stLearn. In the Monocle pseudotime, we observed regional patterns consistent with the Cancer: immune rich and Cancer 1 annotations from the original study (Fig. 5c). However, the spatial noise in the Monocle

**Fig. 4 SpaceFlow reveals evolving cell lineage structures in chicken heart development ST data. a** Annotation of ST spots from the original study[30], where cell types are predicted by mapping scRNA-seq data to ST data. **b** Annotations from SpaceFlow, with identified valve structures colored in blue. **c** The pSM generated by SpaceFlow. **d** The pSM value versus UMAP component 1 from low-dimensional embeddings colored by annotations from SpaceFlow. **e** Hierarchically clustered heatmap of top-3 domain-specific genes for spots in all time points. **f** Gene Ontology (GO) enrichment for the top domain-specific genes (32 genes) in the identified valve structures. Both the color and the length of bars represent the enrichment of GO terms using -log10(p-value) metric from topGO analysis. P values were obtained using the one-sided Fisher's exact test without multiple-testing correction. P values < 0.005 were considered significant. **g** Space-constrained CellChat inferred cell-cell communication of MDK (left) and PTN (right) pathways with signals sent from spots in the valve regions.

pseudotime makes visualizing the overall structure of cancer development difficult. In the pseudotime from stLearn, we can only observe two major types of regions, the Cancer: immune rich regions with larger pseudotime values, and other regions that are more homogeneous but noisy in pseudotime (Fig. 5d). In the SpaceFlow pSM, we see a much clearer representation of the spatiotemporal structure of the cancer cells and the patterns is highly consistent with the expert annotation in Cancer: Immune rich, Cancer 1, and Immune: B/plasma regions (Fig. 5e). The in-situ cancer-1 regions show the lowest pSM values, whereas the Invasive-1 and Invasive-2 regions present the highest pSM values, which indicates the in-situ cancer-1 developmentally preceded than invasive regions. This trajectory can be seen clearly when we plot pSM values against the UMAP component 1 of the embeddings (Fig. 5f). A smooth progression is shown starting from the left bottom corner with the in-situ cancer-1 and branching into APC,B,T-1/2, and in-situ-cancer-2, which then merge into tumor-immune interface populations and end in the invasive-1/2 population. This suggests that in-situ-cancer-2 may be metastasized from in-situ cancer-1.

To study the characteristics of the tumor microenvironment, we identified marker genes for each domain (Fig. 5g). We found invasive-1, invasive-2, and Invasive-Connective (Inva-Conn) share strong expressions of the genes MMP11 and MMP14. Matrix metalloproteinase (MMP) family genes are involved in the breakdown of the extracellular matrix in processes such as metastasis[56]. In the in-situ cancer-1 population, we observed region-specific Interferon-induced expressions, such as IFI27, IFI6, which are associated with cancer growth inhibition and apoptosis promotion[57]. The in-situ cancer-2 population shows a strong and specific expression of TMEM59 and SOX4, which both can promote apoptosis. In tumor-immune interfaces, we found both pro-tumor and anti-tumor gene expressions. For instance, In tumor-immune interface-3, pro-tumor expression markers are TIMP1, a member of MMPs involved in the degradation of the extracellular matrix, whereas IGFBP4, PFDN5, CD63 repress tumor progression[58,59]. We visualized these dual activities of pro-tumor and anti-tumor expression and annotated with pro-tumor or anti-tumor labels to confirm our observations (Fig. 5h). These dual activities are also confirmed in GO analysis (Fig. 5i). The enrichment of the marker genes of tumor-immune interface-1 show pro-tumor GO terms such as: negative regulation of intrinsic apoptotic pathway in response to DNA damage by p53 class mediator, negative regulation of plasmacytoid dendritic cell cytokine production (reduce type I interferon production). Anti-tumor enrichment is also found, such as positive regulation of T cell mediated cytotoxicity (promotes the killing of cancer cells), antigen processing and presentation via MHC class I B (enhances antigen presentation).

To study the cell-cell communication between the invasive (or in-situ cancer) regions and the tumor-immune interfaces, we inferred cell-cell communication through Space-constrained CellChat analysis[50]. We found strong cell-cell communication between the invasive tissue region and the nearby tumor microenvironment through the collagen pathway, which facilities

EMT transition and multiple processes associated with cancer progression and metastasis. Similar cell-cell communication is observed in in-situ cancer, where MDK-SDC1 and APP-CD74 signaling are observed to promote the progression and metastasis (Fig. 5j, k). The detailed function annotations for the communicating ligands and receptors can be found in Supplementary Table 1.

**Discussion**

In this work, we presented SpaceFlow, which (1) encodes the ST data into low-dimensional embeddings reflecting both expression similarity and the spatial proximity of cells in ST data, (2) incorporates spatiotemporal relationships of cells or spots in ST data through a pseudo-Spatiotemporal Map (pSM) derived from the embeddings, and (3) identifies spatial domains with consistent expression patterns, clear boundaries, and less noise.

SpaceFlow achieves competitive segmentation performance with alternative methods when benchmarked against expert annotations. Furthermore, the pSM utilizes the spatially consistent embeddings to reveal pseudo-spatiotemporal patterns in tissue. In DLPFC and Stereo-seq data, the pSM shows layered patterns that are consistent with the developmental sequences of the human cortex and mouse olfactory bulb respectively, which is not visible from non-spatial pseudotime. Applied to chicken heart developmental data, the pSM reveals evolving lineage structures and uncovers the dynamics in the spatiotemporal relationships of cells across different developmental stages, helping to understand the changes of functional and structural organization in tissue development. Studying human breast cancer ST data using SpaceFlow, we demonstrate its potential to identify tumor-immune interfaces and dynamics of cancer progression, providing tools to study tumor evolution and interactions between tumor and the tumor microenvironment.

Though similarity in gene expression and spatial proximity are related in many cases[60], this relationship is not absolute. Pseudotime methods developed for scRNA-seq data, such as Monocle[61] and Slingshot[62] can produce developmental trajectories that are not spatially organized. The pSM developed here can generate spatially contiguous trajectories based on the integrative usage of gene expression and spatial information. Specifically, the spatial regularization in SpaceFlow constrains the low-dimensional embedding spatially so that the embedding is continuous both in space and time. The low-dimensional spatial constraint also reduces noise in the high-dimensional gene expression data resulting in smoother domain segmentation boundaries and spatiotemporal maps.

In practice, the training time of SpaceFlow on ST data with fewer than 10,000 cells is usually less than 5 min on a GPU. The computational cost of training largely depends on the calculation of spatial regularization loss for model optimization, which is quadratic to the number of cells or spots. To accelerate model training, we compute this regularization loss over a random subset of cell-cell pairs (Details in Methods). With a fixed number of cell pairs in the subset, the training can scale linearly with the

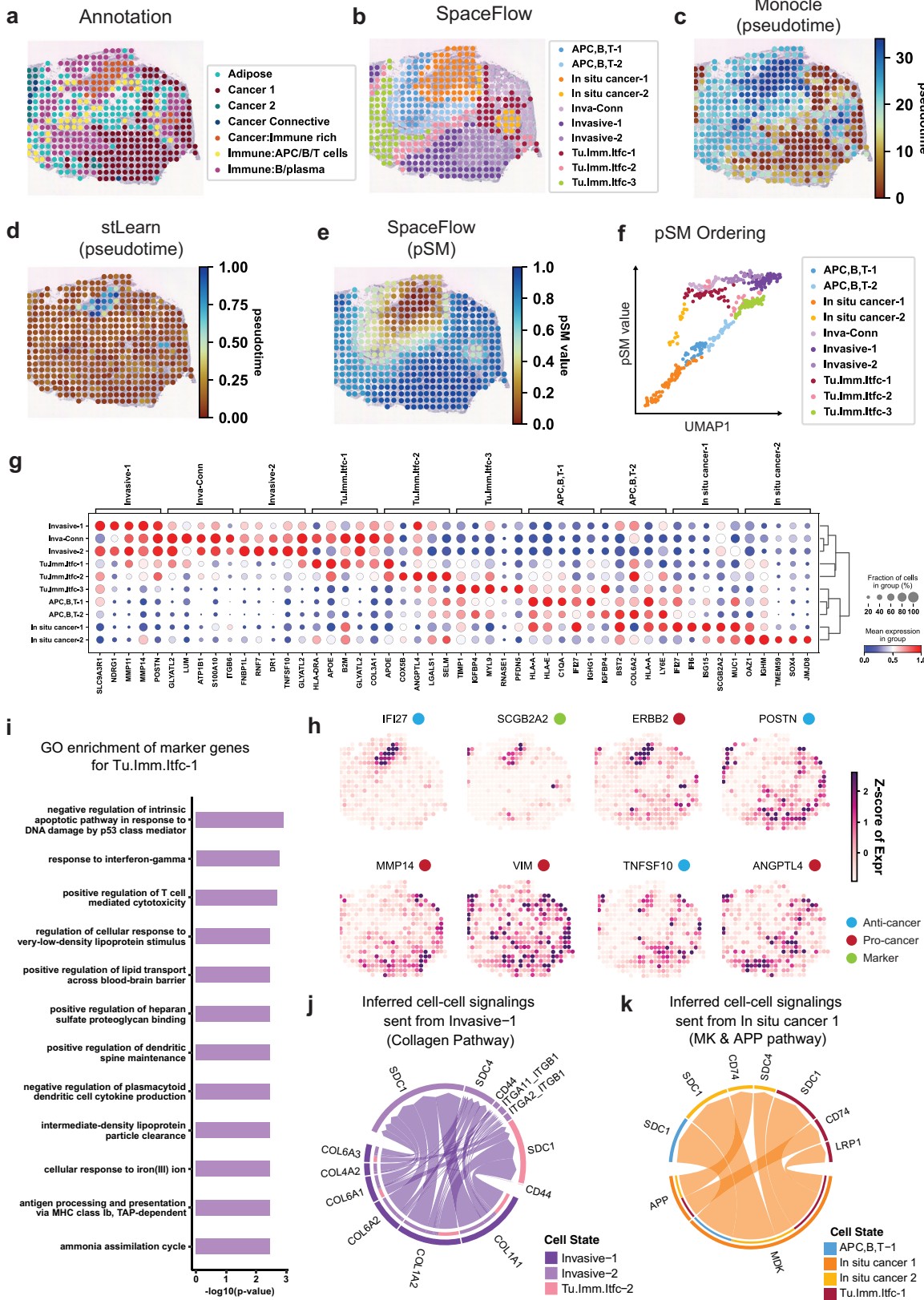

number of cells or spots, and it has been shown not affecting the outcome (Supplementary Fig. 2b–e). In the current implementation, the training will automatically be switched to the approximated regularization strategy when detecting a cell population larger than 10,000. With this strategy, training time varies from 30 s to 3 min for numbers of cells/spots ranging from

3000 to 50,000 on GeForce RTX 2080 Ti GPU. Future work could explore possible alternatives to selecting random subsets such as density-based subsampling[32,33,63], which may be more accurate for estimating the regularization loss.

The spatial regularization used in this work reflect the a priori assumption that nearby cells with similar gene expression are

**Fig. 5 SpaceFlow identifies tumor-immune cell-cell communication in human breast cancer ST data. a** H&E image and annotation from the original study for the spots of sample G in human breast cancer ST data[56]. **b** Domain segmentation from SpaceFlow. **c** Spatial visualization of pseudotime calculated by Monocle. **d** Spatial visualization of pseudotime calculated by stLearn. **e** The pSM from SpaceFlow. **f** The pSM versus UMAP component 1 from low-dimensional embeddings colored by annotations from SpaceFlow. **g** Dot plot of the gene expression of domain-specific markers. The dot size represents the fraction of cells in a domain expressing the marker and the color intensity represents the average expression of the marker in that domain. **h** Spatial expression of top domain marker genes with anti-cancer, pro-cancer, or dual function labels. **i** Gene Ontology (GO) enrichment for the top domain-specific genes (48 genes) in the identified Tumor-Immune-Interface-1 (Tu.Imm.Itfc.1). Enriched GO terms are presented as -log10(p-value) using topGO analysis. *P* values were obtained using the one-sided Fisher's exact test without multiple-testing correction. *P* values < 0.005 were considered significant. Space-constrained CellChat inferred cell-cell communications with cell-cell communications signaling sent from Invasive-1 in Collagen pathway (**j**) and from In-situ cancer 1 in MK & APP pathways (**k**).

more closely related than spatially distant cells with the same level of transcriptional similarity. In connected tissues with low geometric complexity examined in this work, the current spatial regularization with Euclidean distance has good performance. However, it may not cover the complexity of the spatial distribution patterns and dependencies that may vary among different locations of a tissue. Extension of regularization for disjoint tissues like lymph nodes or tissues with high geometric complexity can be developed by location adaptive spatial regularizations. The general framework proposed in SpaceFlow can also be easily extended by combining the latent space regularization with other choices of embedding algorithms, which may offer various tradeoffs in terms of expressive ability and computational efficiency. However, we expect that the general principle that explicit regularization for spatial structure improves performance on ST to hold for a variety of different embedding architectures.

In addition to spatial regularization, SpaceFlow is a flexible framework able to incorporate auxiliary features about connectivity among cells in spatial or single-cell omics data. For example, it can be directly applied to 3D ST data with spatial graph input based on 3D coordinates. Future improvement could be achieved by adapting the framework for spatially resolved Epigenetic data with proper preprocessing steps, such as peak calling on spatially resolved chromatin modification data[64]. Other non-genomic data modalities, such as the local texture features from histological images or expert domain annotation priors could be used to improve the robustness of the SpaceFlow embeddings. Under the SpaceFlow framework, different regularization terms reflect different prior knowledge about the tissue organization and their integration might enhance the performance of the result. In addition, the directed connectivity matrix inferred by RNA velocity[65] could be used as a constraint to derive low-dimensional embeddings consistent with RNA velocity which may improve the representation of developmental trajectory. Overall, SpaceFlow provides a robust framework and an effective tool to incorporate prior knowledge or spatial constraints to ST data analysis for inference of spatiotemporal patterns of cells in tissues.

## Methods

**Data preprocessing**. The raw count expression matrix of ST data is preprocessed as the following. First, genes with expression in fewer than 3 cells and cells with expression of fewer than 100 genes are removed. Next, normalization is performed, where the expression of each gene is divided by total expression in that cell, so that every cell has the same total count after normalization. Then, the normalized expression is multiplied by a scale factor (10,000 by default) and log-transformed with a pseudo-count one. The log-transformed expression matrix of the top 3000 highly variable genes (HVGs) is then selected as the input for constructing the spatial expression graph. We adopt a dispersion-based method to select highly expressed genes[66]. The genes are put into 20 bins based on their mean expression, and then the normalized dispersion is computed as the absolute difference between dispersion (variance/mean) and median dispersion of the expression mean, normalized by the median absolute deviation of each bin. Genes with high dispersion in each bin are then selected.

**Construction of Spatial Expression Graph**. We next convert the log-transformed expression matrix of highly expressed genes into a Spatial Expression Graph (SEG) as the input of our deep graph network. The Spatial Expression Graph is built based on the spatial proximity of cells, with nodes representing cells with expression profiles attached, while edges characterizing the spatial neighborhood of cells. Similarly, in spot-resolution ST data, we use a node to represent a spot in the graph. The SEG is characterized by two matrices, expression matrix $\mathbf{X} = \{x_1, x_2, .., x_n\}$ and spatial adjacency matrix $\mathbf{A} \in \mathbb{R}^{N \times N}$. Here, $x_i$ represents the expression features of the cell or spot $i$, while the element $\mathbf{A}_{i,j}$ in adjacency matrix is equal to 1 if there is an edge between cell/spot i and j, otherwise, $\mathbf{A}_{i,j} = 0$.

We provide two methods for constructing the SEG, namely, alpha-complex-based and k-nearest-neighbor-based. The alpha-complex-based method is used by default, where a Voronoi cell is first created for each cell or spot located at r as:

$$V(r) = \{x \in \mathbb{R}^2 | ||x - r|| \leq ||x - r'||, \forall r' \in C\} \tag{1}$$

where $C$ is the set of coordinates for all the cells or spots, and $||\bullet||$ is the Euclidean distance. Next, the 1-skeleton of the alpha complex[67] is used to determine the neighborhood edges $E$ of the spots, which can be formulated as follows:

$$E = \{(i,j) | \cap_{k \in \{i,j\}} (V(r_k) \cap B(r_k, \delta))\} \tag{2}$$

Where $B(x, \delta)$ is a circle area in $\mathbb{R}^2$ centered at $x$ with a radius $\delta$. The radius $\delta$ is estimated by the mean distance of k nearest neighbors of the spot. In k-nearest-neighbor-based method, the edges of SEG are built based on the top k nearest neighbors of cells.

**Spatially regularized Deep Graph Infomax**. To encode the ST data into low-dimensional embeddings of cells or spots, we use the Deep Graph Infomax (DGI)[38], an unsupervised graph network, as the framework of our model. DGI has the advantage of capturing not only the cell expression patterns but also the cell neighborhood microenvironment, as well as high-level patterns, such as global or regional patterns. Specifically, a two-layer Graph Convolutional Network (GCN) is used as the encoder of DGI with SEG as input. The GCN generates node embeddings $\varepsilon(\mathbf{X}, \mathbf{A}) = \mathbf{H} = \{h_1, h_2, .., h_n\}$ for each cell and spot.

DGI adopts a contrastive learning strategy[68] to learn the encoder, where features are learned through teaching which data points from an unlabeled dataset are similar or distinct. Similar data points are constructed by pairing cell embedding $h_i$ with a global summary vector $s$, whereas the distinct data points are represented by the pairs of the summary vector $s$ and embeddings from a constructed Expression Permuted Graph (EPG). The summary vector $s$ reflects global patterns of SEG, and it is implemented by a sigmoid of the mean of all cell embeddings. EPG is a graph built by random permutating the node features $\mathbf{X}$ in SEG, with the adjacency $\mathbf{A}$ keeping the same. Mathematically, this learning process is achieved by maximizing the following objective function:

$$\text{Loss}_{\text{DGI}} = \frac{1}{2N}(\sum_{i=1}^{N} \mathbb{E}_{(\mathbf{X},\mathbf{A})}[\log D(h_i, s)] + \sum_{j=1}^{N} \mathbb{E}_{(\widetilde{\mathbf{X}},\widetilde{\mathbf{A}})}[\log(1 - D(\widetilde{h}_j, s))]) \tag{3}$$

where $h_i$ is the embedding of node $i$ from the SEG, $\widetilde{h}_j$ is the embedding of node $j$ from the EPG. $\widetilde{\mathbf{X}}$ and $\widetilde{\mathbf{A}}$ are the permuted node features and corresponding adjacency matrix of EPG. The D is the discriminator, which is defined by $D(h_i, s) = \text{Sigmoid}(h_i^T \Theta s)$.

Where $\Theta \in \mathbb{R}^{N_F \times N_F}$ is trainable weight. Through this contrastive learning strategy, the encoder is forced to learn global patterns and neglect random spatial expression patterns in the embeddings.

To enforce the spatial consistency in the embeddings, so that the closeness between embeddings not only reflects the expression similarity but also their spatial proximity, we add a spatial regularization to the objective function in DGI. Mathematically, the revised objective function can be expressed as follows:

$$\text{Loss}_{\text{Total}} = \text{Loss}_{\text{DGI}} + \gamma * \sum_{i=1}^{N} \sum_{j=1}^{N} \frac{\mathbf{D}_{i,j}^{(s)} * (1 - \mathbf{D}_{i,j}^{(z)})}{N * N} \tag{4}$$

where $\mathbf{D}_{i,j}^{(s)}$ represents the spatial distance between cell/spot i to j in Euclidean space, and $\mathbf{D}_{i,j}^{(z)}$ is the embedding distance between cell/spot i to j in embedding space,

N ∗ N is the normalization term, where N is the number of cells or spots in ST data. The spatial regularization penalizes the generation of close embeddings for cells or spots that are spatially far from each other. In another word, so that, the close embeddings caused by the expression similarity are pushed further from each other if they are spatially distant. Strong spatial regularization may overemphasize generation of spatially smooth embeddings which do not necessarily coincide with more textured biological heterogeneity. To mitigate this issue, we added a regularization strength parameter $\gamma$ to control spatial regularization strength relative to the reconstruction loss. The default regularization strength is set to 0.1 as a loose prior for keeping more detailed texture (values ranging from 0 to 1).

**Domain segmentation and pseudo-Spatiotemporal Map**. The domain segmentations are obtained by running Leiden clustering[69] with the low-dimensional embeddings from SpaceFlow as input. By default, the parameter for the local neighborhood size is set to 50 for to produce a smoother segmentation. The pseudo-Spatiotemporal Map (pSM) is calculated by running the diffusion pseudotime (DPT)[31] using the low-dimensional embeddings output from SpaceFlow. The DPT is an algorithm using diffusion-like random walks to estimate the ordering and transitions between cells. Using the embeddings from SpaceFlow that encoded both spatial and expression information of cells as input, DPT can output a spatiotemporal order which is consistent in both space and pseudotime. The root cell for pSM can be specified with prior knowledge, otherwise, in default, the cell that with the largest sum distance to others in embedding space is assigned as the root cell in our strategy.

**Parameters of the model**. The deep graph network is built and trained based on PyTorch. To construct SEG, the default number of nearest neighbors $k$ of a cell or spot for adding edges is set to 15; A larger $k$ will lead to a bigger spatial neighborhood. The DeepGraphInfomax model in PyTorch Geometric library is used for implementing DGI. The default latent dimension size for low-dimensional embeddings is set to 50. A two-layer Graph Convolutional Network (GCN) is utilized as the encoder for SEG with Parametric ReLUs (PReLU) as the activation functions. The number of neurons for both layers is set equal to the low-dimensional embedding size.

**Training procedure**. The optimizer used for training DGI is Adam with a default learning rate $lr = 0.001$ applied[70]. The maximum number of epochs for training is set to 1000, with an early stopping strategy applied to avoid overfitting. Specifically, the minimum epoch for early stopping is set to 100, and the patience of epochs with no loss decrease is set to 50. A GeForce RTX 2080 Ti GPU is used for training the DGI model. The training time varies from 30 s to 3 min numbers of cells/spots ranging from 3000 to 50,000, and the subsampling strategy stated below needed to be applied when the number is greater than 10,000.

**Accelerating the computation of spatial regularization loss**. Because the computational cost of training largely depends on the calculation of spatial regularization loss, which is quadratic to the number of cells or spots, we designed a strategy as follows to accelerating the training. The spatial regularization loss is used in model optimization, which involves calculating the weighted average of an inner product of a spatial distance matrix and an embedding distance matrix. It has $O(M ∗ N^2)$ computational complexity and memory cost, where $N^2$ is the number of edges in a fully-connected spatial graph with $N$ cells, M is the size of the latent dimension. However, during each training step, we compute the spatial regularization loss over a random fixed-size subset of edges, which reduces the computational complexity of regularization loss from quadratic to constant. When tested on the slideseqv2 dataset with 41,876 cells, the computational and memory cost dropped from over 5 h and 18 GB to less than 3 min and 4GB (Supplementary Fig. 4). We additionally found significant improvements in performance applying SpaceFlow to a seqFISH mouse embryogenesis dataset72 (Supplementary Fig. 5).

**Benchmarking**

*Segmentation benchmarking*. To benchmark domain segmentation performance, we compare SpaceFlow against five methods, Seurat 4[39], Giotto[17], stLearn[20], MERINGUE[22], BayesSpace[18] using the LIBD human dorsolateral prefrontal cortex (DLPFC) ST data[40]. To make the domains comparable between benchmarking methods, we set the target number of clusters equal to the number of clusters in annotation for all benchmarking methods. The adjusted Rand index (ARI) is used to quantify the similarity between the clustering result and the annotation.

With Seurat, the RNA transcript counts are used for the input, with genes expressed in fewer than 3 cells filtered, and cells expressing fewer than 100 genes removed. Then, the SCTransform function in Seurat R package is applied to normalize the UMI count data using regularized negative binomial regression. Next, the RunUMAP, FindNeighbors, FindClusters methods are performed on the normalized count data sequentially with the latent dimension size of 50 and default cluster resolution of 0.4.

When benchmarking with stLearn, the count matrix and the spot positions were used as input, which is downloaded directly from the data sources (see Data Availability). The count matrix input was read via Read10X function in the stLearn package. Next, filter_genes, normalize_total, log1p, run_pca functions were applied

sequentially to preprocess data, with the minimal number of genes for filtering set to 3. Next, the histological image of the tissue is preprocessed by using the tiling and extract_features functions. Then, the SME_normalize function is used with the parameter setting of use_data="raw" and weights="physical_distance". Finally, the scale and run_pca are performed on the normalized data with number of principal components of 50. The principal components from normalized data will then be used for segmentation or pseudotime analysis via Leiden and DPT, respectively.

With Giotto, we input the count matrix and the spot positions, and then applied the normalizeGiotto, addStatistics, calculateHVG to preprocess data and identify highly variable genes (HVG). HVGs expressed in at least 3 cells and with a mean normalized expression greater than 0.4 are then feed into runPCA function for the principal components. The spatial network was then created through the createSpatialNetwork function with the parameter for the kNN method set to $k = 5$ and a maximum distance of 400 in kNN. Finally, the doHMRF method is used for clustering with the parameter beta set to 40.

For BayesSpace benchmarking, we input expression matrix and the spot positions through the getRDS("2020_maynard_prefrontal-cortex") method. Next, the modelGeneVar and getTopHVGs methods in scran method are used to model the variance of log-expression profile of each gene and extract the top 2000 highly variable genes. Then, the runPCA function in the scater package is used for principal components. The BayesSpace clustering method spatialCluster is applied with 15 principal components, with 50,000 MCMC iterations and gamma = 3 for smoothing.

For MERINGUE benchmarking, we input the spatial locations and the top 50 principal components from the expression matrix of ST data. Next, the spatial adjacency weight matrix is constructed using the getSpatialNeighbors function in the R package of MERINGUE, with a setting of filterDist = 2. Then, getSpatiallyInformedClusters is performed to get spatially informed clusters by weighting graph-based clustering with spatial information, with a setting of k = 20, alpha = 1, beta = 1.

*Pseudo-spatiotemporal map benchmarking*. The pSM is compared to the spatial embedding method stLearn[20], and three non-spatial pseudotime methods, Seurat 4[39], Monocle[61], and Slingshot[62]. In stLearn, because the histological image of the tissue is required for spatial-aware embedding, we only made comparisons when the histological was available. We calculated pseudotime from stLearn by running the diffusion pseudotime (DPT) (Haghverdi et al. 2016) using the stLearn embedding. In Seurat 4, the DPT is run using the principal components of the expression data, whereas in Monocle and Slingshot, the recommended workflows with the default parameters are performed.

**Downstream analysis**

*Marker genes identification*. To identify marker genes that can best characterize specific expressions for domains output from SpaceFlow, the rank_genes_groups method in the Scanpy package (v1.8.2) is used. When performing this method, the Wilcoxon rank-sum test with a Benjamini–Hochberg $p$ value correction is applied. The cutoff of the adjusted $p$ value for domain-specific marker genes is set to 0.01.

*Domain annotation*. The domains identified by SpaceFlow are annotated based on the literature report of the domain-specific marker genes. The details of the literature support and marker gene list can be found in Supplementary Data 1 and 2.

*Gene Ontology enrichment analysis*. The Gene Ontology (GO) Enrichment Analysis in the GO Consortium website is carried out to identify the enriched GO terms for domain-specific maker genes with adjusted $p$ value < 0.01.

*Space-constrained CellChat analysis*. CellChat analysis is performed on ST data using the domain labels from SpaceFlow as groupings. Inferred CellChat communications between domains are further scrutinized such that the communication links are only allowed between spatially adjacent domains. The CellChat v1.1.3 is used under a R v4.1.2 environment.

**Reporting summary**. Further information on research design is available in the Nature Research Reporting Summary linked to this article.

## Data availability

All data analyzed in this paper can be downloaded in raw form from the original publication. Specifically, the DLPFC data is available in the "spatialLIBD package [http://spatial.libd.org/spatialLIBD]". The processed Stereo-seq data from mouse olfactory bulb tissue is accessible at "SEDR analyses [https://github.com/JinmiaoChenLab/SEDR_analyses]". The chicken heart ST data is retrieved from GEO database under accession code "GSE149457 [https://www.ncbi.nlm.nih.gov/geo/query/acc.cgi?acc=GSE149457]". The human breast cancer ST data can be obtained from the Zenodo dataset "4751624". The sample used is the same as the one demonstrated in the original paper (patient G-sample 1). Both chicken heart ST data and breast cancer ST data were sequenced by 10x Visium platform. The Slide-seq V2 can be accessed in Squidpy package[71] or

downloaded from "Broad Institute database [https://singlecell.broadinstitute.org/single_cell/study/SCP815/highly-sensitive-spatial-transcriptomics-at-near-cellular-resolution-with-slide-seqv2]". The seqFISH data can be accessed at the "Spatial Mouse Atlas [https://marionlab.cruk.cam.ac.uk/SpatialMouseAtlas/]". The Gene Ontology Consortium database can be accessed via "Gene Ontology Consortium [http://geneontology.org/]". All other relevant data supporting the key findings of this study are available within the article and its Supplementary Information files or from the corresponding author upon reasonable request.

## Code availability

The SpaceFlow package is implemented in Python with a dependency of Pytorch and is available on the GitHub repository "SpaceFlow [https://github.com/hongleir/SpaceFlow]". It is also deposited at Zenodo dataset "6668286".

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

## Acknowledgements
The project was partly supported by the National Science Foundation grant DMS176372, the National Institutes of Health grants U01AR073159, R01DE030565, and P30AR075047, and a Simons Foundation Grant (594598).

## Author contributions
Z.C., Q.N., H.R., and B.W. conceived the project; H.R. implemented the algorithm and code, and conducted data analysis. All the authors wrote and approved the manuscript; Q.N. supervised the research.

## Competing interests
The authors declare no competing interests.
