## [Peer Review File · Nature Communications]

REVIEWER COMMENTS

Reviewer #1 (Remarks to the Author):

The paper describes a graph convolutional network approach towards analysis of spatially resolved, developmental biology data. A tissue graph is constructed, with cells as nodes (with transcriptomes attached) and edges are spatial adjacency relations. A standard deep graph infomax framework is used to train an encoder for spatial expression patterns. This is followed by 1) a clustering (segmentation) of spatial patterns, and a gene expression embedding that enables a pseudo-time ordering, and as such this work is simultaneously solving lineage tracing as well as spatial pattern discovery. It is one of the first efforts that I ran into that tries to address both issues at the same time in the context of single-cell spatial transcriptomics.

I do have a number of concerns, that need to be addressed though:

- I find the introduction and related work discussion too much oriented towards only discussing spatial transcriptomics prior work, while several of the issues (for instance spatial information integrations in high-dimensional data embedding methods) have been addressed in adjacent domains; I'm from a generic data visualization world, where single-cell biology is viewed as "just an application". There's a plethora of work on spatially-aware embedding methods for high-dimensional image data (e.g. imaging mass cytometry & satellite imagery (for instance the work of Vieth e.a. at Pacific Vis [https://graphics.tudelft.nl/Publications-new/2022/VVLEH22/.](https://graphics.tudelft.nl/Publications-new/2022/VVLEH22/)), Imaging Mass Spectrometry (Baluff, 2016, Analytical Chemistry), several works of Abdelmoula (including his work on spatially mapped tSNE in PNAS 2016, NatComms 2022).

In the current introduction on related work has a bit too much of a "reinventing the wheel" flavor, and should be extended with an awareness of similar approaches outside of the relatively young "new kid on the block" field of spatial transcriptomics; Also, these prior methods suffer much less from the limited scalability of the proposed method, and should therefore be discussed in either intro or discussion. Finally, a lot of really nice work has been described on bulk spatial transcriptomics data, such as the (developing) Allen mouse and human brain atlases (for instance Bohland's work, or Huisman's dual-tSNE work in NAR (2017)). This is "spatial transcriptomics analysis avant la lettre", but many of the methods developed for these resources apply one on one to the data under investigation here...

- The work is indeed very attractive in that it solves the pseudotime and spatially-aware embedding at the same time; What I miss though, is the "why" behind the choice of a GCN approach, and the "why not something else, more efficient"; It should at least be discussed why this would not have been possible with state-of-the-art other data, ultrafast and scalable embedding methods, that suffer less from scalability problems (such as Hierarchical SNE or Hierarchical UMAP); For instance a dual embedding strategy of sample / feature embedding would likely yield similar results (for instance as in Pezzotti's WAOWViz... So a GCN-based approach is fine, but it should be explained why, especially due to the combinatorial explosion of the compute time...

- I find the description of how the temporal ordering is achieved a bit sketchy... Some random-walk-based distance between the gene clusterings? Please elaborate a bit further how this distance is computed...
- The authors choose to integrate the spatial information in a distance term in the loss function... However, it's not motivated why the distance? Why not some kind of local texture features? I think a distance term may overemphasize generation of nice contiguous patches, which do not necessarily coincides with more textured biological heterogeneity... Please elaborate on the choice of the local distance, and why not some other local image properties (SIFT features?)
- To alleviate the scaling problem, the authors describe a subsampling strategy based on sparsifying the graph using random subsampling; However, random subsampling has been known to discard rare cell types, such that they can be overlooked in downstream analyses; This sounds undesirable and counterintuitive: first spend a lot to acquire super detailed spatially resolved cell phenotyping, and the discard them again because the processing is computationally too expensive? Especially because one of the big perks of single-cell analyses is the discovery of rare cell types that can be causal for disease and as such, why didn't the authors opt for density based subsampling, such as proposed in hierarchical data embedding methods such as HSNE and HUMAP. These have been shown to preserve rare cell types (Van Unen, NatComms 2017), since similarities are still computed on the full data set...

I find the validations and comparisons solid, and indeed the method seems to perform better than many other methods. The uncovered biology seems to confirm logical patterns, and prior knowledge, and as such, can certainly be useful in gaining further insight in this complex data.

Alltogether, I think the paper is interesting, and has merit for NatComms, provided that:

- the introduction is broadened as indicated with a much broader state-of-the-art awareness outside of spatial transcriptomics.
- the "why" of the choices (and the "why not" of obvious other choices like fast data embedding methods) should be better motivated
- Some more detail / background on a few aspects of the method.

Reviewer #2 (Remarks to the Author):

In this manuscript authors propose to solve an important problem of learning spatially-consistent low-dimensional embeddings using gene expression and spatial information. There are published tools to solve similar problem but the novelty here comes from learning spatio-temporal order which is consistent in both space and time. Overall the manuscript is well written but I have a few concerns:

* Figure 2d, it's not clear what author's are suggesting through this figure, specially the gradient in the UMAPs are not consistent with the defined layers.

* Figure 2g, authors mention that "the spatially specific expression in the identified domain 3 may be related to long-term synaptic activity" through GO term analysis, but it's not clear if it's expected ? Without prior knowledge it's not clear the usefulness of Figure 2g.

* Figure 3b, It's very hard to interpret pseudotime in the context of spatial data but according to the author's intuition the developmental sequence of the layers should be consistent with the pseudotime. Overall this result is great but strangely the green color from the pseudotime occurs multiple times (outer and inner rims of red / EPL layer), do authors have a sense why ?

* Figure 3d looks fantastic !

* MERINGUE (Miller, Brendan F., et al. "Characterizing spatial gene expression heterogeneity in spatially resolved single-cell transcriptomic data with nonuniform cellular densities." *Genome research* 31.10 (2021): 1843-1855.) is another tool which author's can consider benchmarking against but I highly recommend citing the paper as it solves the similar problem.

* Figure 3, author's did a great job in comparing Spaceflow with methods using only spatial information or only pseudotime information but I think one type of comparison is missing i.e. It's highly probable if we run Monocle on BayesSpace (or other tools like Giotto or MERINGUE) defined embedding the learned pseudotime would be very similar to the one shown for SpaceFlow in Figure 3a. I think it'd be interesting to make such comparison where one can combine separate off-the-shelf methods to learn spatio-temporal order compared to Spaceflow.

* Comparatively to the rest of the paper the text around Figure 4 is a bit disappointing, there are a lot missing pieces. For example D4 and D7 data, what does each blob mean ? does even make sense to have pseudotime in multiple similar tissues ?

* Figure 5, Again I think the comparison here should be BayesSpace (or other methods like MERINGUE/)+Monocle, Monocle alone is not a fair comparison. The point about Spaceflow being better than Monocle is already made in earlier figure.

* Currently Seurat does not perform spatially aware clustering nor does it do pseudotime analysis (uses Monocle) on its own. Looking at Figure 2/3, it seems a bit unfair comparing methods with "extra" spatial information with unsupervised analysis.

Reviewer #3 (Remarks to the Author):

The authors presented SpaceFlow, a method that learns latent space embeddings of cells (or spots) combining both the spatial locations of cells (or spots) and the gene expression profiles of the cells (or spots). The learned integrated embedding can be used to perform domain segmentation (through applying clustering methods to the embedding) or to learn pseudotime of the cells (or spots) by applying diffusion pseudotime (DPT) method. The method has been applied to multiple datasets and led to observations consistent with known knowledge. The method is novel and the pipeline can be used to find spatial-constrained development dynamics which can potentially make this work impactful. However, I have some concerns regarding the current form of the manuscript.

Major concerns:

1. Regarding the results on the 10x visium human cortex data: when Seurat v4 was used to generate Figs. 2a-d, was spatial location information used or was only the gene expression information used? Also, every baseline method may involve some parameter settings. Details on input and parameter settings when running baseline methods including Seurat v4, stLearn, Giotto, SpaGCN and BayesSpace should be provided.
2. Fig. 2f shows the expression patterns of 5 selected genes. How are these genes selected? Is each gene the top 1 domain-specific gene for each domain? Do these genes have annotated functions in databases or literature which support their domain-specific property?
3. In Figure 3, DPT was applied to SpaceFlow embedding to generate the pSM, and the inferred pseudotime is compared with Seurat, Monocle and Slingshot. It's not clear whether the spatial information is used for Seurat in this analysis (please clarify), but for Monocle and Slingshot no spatial information was used. While this shows the advantage of incorporating spatial information, it doesn't show the advantage of SpaceFlow over methods like stLearn. Since DPT can be applied to almost any embedding to generate pseudotime, one can run stLearn to obtain an embedding using both spatial and gene expression information, and then apply DPT. Therefore, for the comparisons in Figs. 3a-b, I suggest that the authors also compare with the pseudotime obtained by stLearn followed by DPT.

4. In Lines 271-272, the authors mention “By contrast, the identified valve structures show light or dark blue color in the pSM from D7 to D14, suggesting the ordering is relatively late compared with the regions colored in red, yellow, and green. By plotting pSM values of spots against the first component of the UMAP embedding, one can better observe these patterns (Fig. 4d).” It’s hard for me to observe the above mentioned pattern in Fig. 4d. Please elaborate.

5. For Fig. 3d, please specify the input and parameter setting for Seurat and Scanpy. Particularly, do Seurat and Scanpy use spatial information? The discussion in Lines 228-232 the authors mentioned that Seurat segmentation does not have a high resolution, and it combines regions that could be separated by SpaceFlow. Did the users try increasing the “resolution” parameters in Seurat and see whether it helps Seurat to find regions with higher resolution?

6. Line 497-499: “The neighborhood graph is constructed based on UMAP of the embeddings”: does it mean the neighborhood graph is constructed using the 2-d information from UMAP? What are the reasons for not using the embeddings from spaceflow to construct the neighborhood graph?

Minor points:

In Fig. 2e, the row and column labels are numbers 0 to 4 - what do the numbers mean?

Which dataset is used in Fig. 3a? One would guess it’s the same 10x visium human cortex data used for Fig. 2, but it needs to be specified for Fig. 3a.

Line 301: SCD is used here but it seems what it stands for is not defined.

Fig. 4e: the labels can’t be read clearly. Please use a figure with higher resolution.

The manuscript has some typos which need to be corrected and confusing notations which need to be clarified.

1. Line 118: please check grammar in this sentence “need to be far from...”?

2. Line 430: HVG means highly variable genes but the sentence used highly expressed genes for the full form. Please clarify.

3. Line 451: there should be a space after “and”

4. 468-478: some notations are underlined. I don't think it is well-known what it means to underline a notation – please explain.

5. 532-533: the notation n^2 is not formal.

Response to Reviewer #1

The paper describes a graph convolutional network approach towards analysis of spatially resolved, developmental biology data. A tissue graph is constructed, with cells as nodes (with transcriptomes attached) and edges are spatial adjacency relations. A standard deep graph infomax framework is used to train an encoder for spatial expression patterns. This is followed by 1) a clustering (segmentation) of spatial patterns, and a gene expression embedding that enables a pseudo-time ordering, and as such this work is simultaneously solving lineage tracing as well as spatial pattern discovery. It is one of the first efforts that I ran into that tries to address both issues at the same time in the context of single-cell spatial transcriptomics. I do have a number of concerns, that need to be addressed though:

Response: We thank the reviewer for the insightful comments. These comments are valuable and helpful for improving our work. We've have made substantial changes that highlighted in red in the revised manuscript to address those points. Point-to-point responses for the reviewer's comments are provided below.

R1-1. *I find the introduction and related work discussion too much oriented towards only discussing spatial transcriptomics prior work, while several of the issues (for instance spatial information integrations in high-dimensional data embedding methods) have been addressed in adjacent domains; I'm from a generic data visualization world, where single-cell biology is viewed as "just an application". There's a plethora of work on spatially-aware embedding methods for high-dimensional image data (e.g. imaging mass cytometry & satellite imagery (for instance the work of Vieth e.a. at Pacific Vis*

[https://graphics.tudelft.nl/Publications-new/2022/VVLEH22/.](https://graphics.tudelft.nl/Publications-new/2022/VVLEH22/)),

[https://graphics.tudelft.nl/Publications-new/2022/VVLEH22/.](https://graphics.tudelft.nl/Publications-new/2022/VVLEH22/)), *Imaging Mass Spectrometry* (Baluff, 2016, *Analytical Chemistry*), several works of Abdelmoula (including his work on spatially mapped tSNE in *PNAS* 2016, *NatComms* 2022).

*In the current introduction on related work has a bit too much of a “reinventing the wheel” flavor, and should be extended with an awareness of similar approaches outside of the relatively young “new kid on the block” field of spatial transcriptomics; Also, these prior methods suffer much less from the limited scalability of the proposed method, and should therefore be discussed in either intro or discussion. Finally, a lot of really nice work has been described on bulk spatial transcriptomics data, such as the (developing) Allen mouse and human brain atlases (for instance Bohland’s work, or Huisman’s dual-tSNE work in *NAR* (2017)). This is “spatial transcriptomics analysis avant la lettre”, but many of the methods developed for these resources apply one on one to the data under investigation here...*

Response: We thank the reviewer for this helpful advice for positioning our work, and spatial transcriptomics, within the broader field of relevant research. In the revision, we have expanded Introduction (Line 42-54 Page 2) to include more the relevant background and discuss how spatial transcriptomics fits in to broader areas of study.

R1-2. *The work is indeed very attractive in that it solves the pseudotime and spatially-aware embedding at the same time; What I miss though, is the “why” behind the choice of a GCN approach, and the “why not something else, more efficient”; It should at least be discussed why this would not have been possible with state-of-the-art other data,*

ultrafast and scalable embedding methods, that suffer less from scalability problems (such as Hierarchical SNE or Hierarchical UMAP); For instance a dual embedding strategy of sample / feature embedding would likely yield similar results (for instance as in Pezzotti's WAOWViz... So, a GCN-based approach is fine, but it should be explained why, especially due to the combinatorial explosion of the compute time...

Response: We thank the reviewer for helpful comments on additional context for our work. In the revision, we have expanded Introduction (Line 86-96 Page 4) with additional references added and more details in the Overview of SpaceFlow in Results (Line 140-143 Page 6) to more clearly present methodologies for embeddings and the reasons why we chose a GCN architecture, as well as the choice of DGI as a GCN architecture. We have also addressed the question of scalability of our GCN method in responding your question R1-5.

R1-3. *I find the description of how the temporal ordering is achieved a bit sketchy... Some random-walk-based distance between the gene clusterings? Please elaborate a bit further how this distance is computed...*

Response: We thank the reviewer for pointing out the room to improve clarity on the pseudotime method. In the revision, we have expanded the description of calculation of the pseudo-Spatiotemporal Map (pSM) in the Methods section (Line 581-588 Page 23).

R1-4. *The authors choose to integrate the spatial information in a distance term in the loss function... However, it's not motivated why the distance? Why not some kind of local texture features? I think a distance term may overemphasize generation of nice*

contiguous patches, which do not necessarily coincides with more textured biological heterogeneity... Please elaborate on the choice of the local distance, and why not some other local image properties (SIFT features?)

Response: We thank the reviewer for this comment. To better explain the choice of spatial regularization in our method, in the revision we've expanded the Overview of SpaceFlow in the Results section (Line 145-150 Page 6), as well as discussions about other possibilities for features to use in regularization in the discussion section (Line 469-480, Line 487-491 Page 19). To address the possibility of overemphasizing spatial smoothness, we have added a regularization strength parameter which users can control with their prior knowledge or for their own purpose (Line 570-575 Page 23).

R1-5. *To alleviate the scaling problem, the authors describe a subsampling strategy based on sparsifying the graph using random subsampling; However, random subsampling has been known to discard rare cell types, such that they can be overlooked in downstream analyses; This sounds undesirable and counterintuitive: first spend a lot to acquire super detailed spatially resolved cell phenotyping, and the discard them again because the processing is computationally too expensive? Especially because one of the big perks of single-cell analyses is the discovery of rare cell types that can be causal for disease and as such, why didn't the authors opt for density-based subsampling, such as proposed in hierarchical data embedding methods such as HSNE and HUMAP. These have been shown to preserve rare cell types (Van Unen, NatComms 2017), since similarities are still computed on the full data set...*

Response: We thank the reviewer for their very important note on subsampling. In this work, the described subsampling refers only which spots are used in the computation of the spatial regularization loss (as described in Methods Line 607-618 Page 24). No cells or spots are ever discarded, so we will not lose rare cell types. To improve the clarity on this point, in the revision we've added new paragraphs in both Discussion (Line 455-467 Page 18) and Methods (Line 607-618 Page 24).

Response to Reviewer #2

In this manuscript authors propose to solve an important problem of learning spatially-consistent low-dimensional embeddings using gene expression and spatial information. There are published tools to solve similar problem but the novelty here comes from learning spatio-temporal order which is consistent in both space and time. Overall, the manuscript is well written but I have a few concerns:

Response:

We thank the reviewer for their appreciation of the value of SpaceFlow method and for the insightful comments. Detailed responses to the individual comments are listed below.

R2-1. *Figure 2d, it's not clear what author's are suggesting through this figure, specially the gradient in the UMAPs are not consistent with the defined layers.*

Response: We apologize for the confusion in this visualization. In the revision, we have revised the Figure 2d (Page 31) for a more straightforward visualization on the meaning of the color gradient, which is the distance to the origin for each cell. The caption of Figure 2d is also updated at Line 754-756 on Page 32. In order to clarify the interpretation of this figure, we have adjusted the Results section accordingly (Line 206-208 Page 9).

R2-2. *Figure 2g, authors mention that "the spatially specific expression in the identified domain 3 may be related to long-term synaptic activity" through GO term analysis, but it's not clear if it's expected? Without prior knowledge it's not clear the usefulness of Figure 2g.*

Response: We thank the reviewer for this comment. To better address the significance, in the revision we have added literature support on some marker genes, as well as more

details on the significance and interpretation of our findings in the Results (Line 229-236 Page 9).

R2-3. *Figure 3b, it's very hard to interpret pseudotime in the context of spatial data but according to the author's intuition the developmental sequence of the layers should be consistent with the pseudotime. Overall, this result is great but strangely the green color from the pseudotime occurs multiple times (outer and inner rims of red / EPL layer), do authors have a sense why?*

Response: We thank the reviewer for their careful observation. As pseudotime values reflect distance from a chosen root cell, it is possible for distinct tissues that “sandwich” the root cell on either side to have similar pseudotime values. Additionally, the spatial pseudotime of SpaceFlow in Figure3b is found to be consistent with the biologically observed sequence of the layers in mouse olfactory bulb. We’ve revised the text in the revision to make this clear (Line 261-264 Page 11).

R2-4. *Figure 3d looks fantastic!*

Response:

Thank you for the positive feedback.

R2-5. *MERINGUE (Miller, Brendan F., et al. "Characterizing spatial gene expression heterogeneity in spatially resolved single-cell transcriptomic data with nonuniform cellular densities." *Genome research* 31.10 (2021): 1843-1855.) is another tool which author's can consider benchmarking against but I highly recommend citing the paper as it solves the similar problem.*

Response: Thank you for the helpful suggestion on expanding our benchmarking. In the revision we have added MERINGUE benchmarking results to Figure 2a, b, Figure 3d, SI Figure 1, 4, 5 along with an expansion on relevant main texts in Results (Line 248-250 Page 10, Line 375-377, Line 380-385 Page 15). Please see the detailed response with respect to each figure below.

R2-6. *Figure 3, author's did a great job in comparing Spaceflow with methods using only spatial information or only pseudotime information but I think one type of comparison is missing i.e. It's highly probable if we run Monocle on BayesSpace (or other tools like Giotto or MERINGUE) defined embedding the learned pseudotime would be very similar to the one shown for SpaceFlow in Figure 3a. I think it'd be interesting to make such comparison where one can combine separate off-the-shelf methods to learn spatio-temporal order compared to Spaceflow.*

Response: We thank reviewer for the insightful comment. We've added the suggested analysis in Figure3a using the embeddings from stLearn followed with the diffusion pseudotime (DPT). The corresponding text for elaborating the result of Figure 3a is also updated (Line 248-250 Page 10). The reason we use stLearn and DPT is twofold. First, not all methods (such as BayesSpace, Giotto, SpaGCN, MERINGUE) can generate embeddings to use for pseudotime analysis. Second, pseudotime methods such as Monocle and Slingshot cannot take embedding as input. In addition, stLearn is unable to handle the Stereo-seq data in Figure 3b, because it cannot generate spatially normalized embeddings without the histological image. In the revision we added the explanations in Methods (Line 673-676 Page 27).

R2-7. Comparatively to the rest of the paper the text around Figure 4 is a bit disappointing, there are a lot missing pieces. For example D4 and D7 data, what does each blob mean? does even make sense to have pseudotime in multiple similar tissues?

Response: We thank the reviewer for the comment and apologize for the incomplete explanations. To improve the clarity, in the revision we've expanded the description of result for Figure 4 (Line 297-299 Page 12). We agree with the reviewer that comparing pseudo-time across tissues with different time points may not make sense, and so we only compared and explained the pseudotime ordering within each tissue section separately for each time point. We have added a sentence to clarify this in the results section (Line 315-316 Page 13).

R2-8. Figure 5, Again I think the comparison here should be BayesSpace (or other methods like MERINGUE) + Monocle, Monocle alone is not a fair comparison. The point about Spaceflow being better than Monocle is already made in earlier figure.

Response: We thank the reviewer for the comment on improving the significance of the comparison. In the revision we have added the results from stLearn+DPT in Figure 5d with the revision in text (Line 375-377, Line 380-385 Page 15), showing that Spaceflow has better performance compared to both spatial and non-spatial alternative methods.

R2-9. Currently Seurat does not perform spatially aware clustering nor does it do pseudotime analysis (uses Monocle) on it's own. Looking at Figure 2/3, it seems a bit unfair comparing methods with "extra" spatial information with unsupervised analysis.

Response: We agree to the reviewer that a fair comparison would be against spatial-based methods. In the revision, we have added the segmentation result using MERINGUE in Figure 2a, b, Figure 3d, SI Figure 1, 4, 5, and the stLearn pseudotime result in Figure 3a, Figure 5d, and updated the corresponding text in the revision (Line 248-250 Page 10, Line 375-377, Line 380-385 Page 15).

Response to Reviewer #3

The authors presented SpaceFlow, a method that learns latent space embeddings of cells (or spots) combining both the spatial locations of cells (or spots) and the gene expression profiles of the cells (or spots). The learned integrated embedding can be used to perform domain segmentation (through applying clustering methods to the embedding) or to learn pseudotime of the cells (or spots) by applying diffusion pseudotime (DPT) method. The method has been applied to multiple datasets and led to observations consistent with known knowledge. The method is novel and the pipeline can be used to find spatial-constrained development dynamics which can potentially make this work impactful. However, I have some concerns regarding the current form of the manuscript.

Response: We thank the reviewer for the appreciation of the novelty of SpaceFlow and the insightful comments to strengthen the method. Substantial revisions have been made and are highlighted with red throughout the updated manuscript. Our detailed responses to each comment are provided below.

R3-1. *Regarding the results on the 10x visium human cortex data: when Seurat v4 was used to generate Figs. 2a-d, was spatial location information used or was only the gene expression information used? Also, every baseline method may involve some parameter settings. Details on input and parameter settings when running baseline methods including Seurat v4, stLearn, Giotto, SpaGCN and BayesSpace should be provided.*

Response: We thank the reviewer for the suggestions on improving clarity of technical details. We've supplied the spatial or non-spatial method information for the Figs. 2a-d in

the revision (Line 162-163 Page 7). In addition, we have added details of parameters and functions used in benchmarking (Line 630-668, Page 25-27).

R3-2. *Fig. 2f shows the expression patterns of 5 selected genes. How are these genes selected? Is each gene the top 1 domain-specific gene for each domain? Do these genes have annotated functions in databases or literature which support their domain-specific property?*

Response: We thank the reviewer for this comment. These five genes are the top-1 domain-specific genes. In the revision we have added text for clarifying this Fig. 2f caption (Line 759 Page 32), and provided additional literature support on our findings in Results (Line 220-224 Page 9).

R3-3. *In Figure 3, DPT was applied to SpaceFlow embedding to generate the pSM, and the inferred pseudotime is compared with Seurat, Monocle and Slingshot. It's not clear whether the spatial information is used for Seurat in this analysis (please clarify), but for Monocle and Slingshot no spatial information was used. While this shows the advantage of incorporating spatial information, it doesn't show the advantage of SpaceFlow over methods like stLearn. Since DPT can be applied to almost any embedding to generate pseudotime, one can run stLearn to obtain an embedding using both spatial and gene expression information, and then apply DPT. Therefore, for the comparisons in Figs. 3a-b, I suggest that the authors also compare with the pseudotime obtained by stLearn followed by DPT.*

Response: We agree with the reviewer that the comparison against spatial-based methods is needed. In the revision, we've added the stLearn + DPT result in Figure 3a and adjusted the texts accordingly (Line 248-250 Page 10). However, we found stLearn is unable to handle Stereo-seq data in Figure 3b because its spatial embedding algorithm is based on applying histological information which is not available in this dataset. We added this explanation in Methods (Line 673-676 Page 27).

R3-4. *In Lines 271-272, the authors mention “By contrast, the identified valve structures show light or dark blue color in the pSM from D7 to D14, suggesting the ordering is relatively late compared with the regions colored in red, yellow, and green. By plotting pSM values of spots against the first component of the UMAP embedding, one can better observe these patterns (Fig. 4d).” It’s hard for me to observe the above mentioned pattern in Fig. 4d. Please elaborate.*

Response: We thank the reviewer for the comment. In the revision we have revised and expanded this description (Line 326-328 Page 13) with the following explanation, “By plotting pSM values of spots against the first component of the UMAP embedding (Fig. 4d), similar patterns is observed, where the cells with valve annotations colored in blue shows intermediate pSM values (y-axis) and lies in the middle of the trajectories in Fig. 4d.”

R3-5. *For Fig. 3d, please specify the input and parameter setting for Seurat and Scanpy. Particularly, do Seurat and Scanpy use spatial information? The discussion in Lines 228-232 the authors mentioned that Seurat segmentation does not have a high resolution,*

and it combines regions that could be separated by SpaceFlow. Did the users try increasing the “resolution” parameters in Seurat and see whether it helps Seurat to find regions with higher resolution?

Response: We thank the reviewer for the detailed comment on improving technical clarity and addressing resolution. In the revision, we made the clarification about whether spatial information is used in Fig.3d (Line 277-278 Page 11). We also have added the result from spatial clustering method MERINGUE (Miller, Brendan F., et al.) in Fig.3d for an additional benchmarking comparison. Parameters and functions used are now listed in Methods (Line 630-668, Page 25-27). We have also included additional results for Seurat with increased resolutions in SI Fig. 2a.

R3-6. *Line 497-499: “The neighborhood graph is constructed based on UMAP of the embeddings”: does it mean the neighborhood graph is constructed using the 2-d information from UMAP? What are the reasons for not using the embeddings from spaceflow to construct the neighborhood graph?*

Response: We apologize for the confusion, which was an error in the manuscript. The neighborhood graph of cells was computed using the embeddings from SpaceFlow, not the UMAP of the embedding. The Leiden graph-clustering method was then used to cluster cells using the neighborhood graph directly. We have revised the text (Line 578-580 Page 23) in the manuscript to make this clear.

R3-7. *In Fig. 2e, the row and column labels are numbers 0 to 4 - what do the numbers mean?*

Response: We thank the reviewer for this comment. We have added the label of “segmentation” to Figure 2e in the revision (Page 31).

R3-8. *Which dataset is used in Fig. 3a? One would guess it's the same 10x visium human cortex data used for Fig. 2, but it needs to be specified for Fig. 3a.*

Response: We thank the reviewer for this note on improving clarity. In the revision we have added additional description about the dataset in the caption of Figure3a (Line 769 Page 34), which explains that the DLPFC dataset is the same dataset as in that in Figure 2.

R3-9. *Line 301: SCD is used here but it seems what it stands for is not defined.*

Response: We thank the reviewer for noticing this error. We've changed the typo of SCD into SDC2 with an update for literature support in the revision (Line 356-357 Page 14).

R3-10. *Fig. 4e: the labels can't be read clearly. Please use a figure with higher resolution.*

Response: We have replaced all figures with much higher resolution versions in the revision.

R3-11.

Line 118: please check grammar in this sentence “need to be far from...”?

Line 430: HVG means highly variable genes but the sentence used highly expressed genes for the full form. Please clarify.

Line 451: there should be a space after “and”.

Line 468-478: *some notations are underlined. I don't think it is well-known what it means to underline a notation – please explain.*

Line 532-533: *the notation n^2 is not formal.*

Response: We thank the reviewer for the careful reading of our manuscript. In the revision, we have made changes accordingly to correct and clarify each of these points.

REVIEWERS' COMMENTS

Reviewer #1 (Remarks to the Author):

The authors went through great lengths to address all the points I raised, and I think, also based on the other reviewers' comments, the manuscript has much improved. As such, I believe the manuscript can now be accepted as is.

Reviewer #2 (Remarks to the Author):

All my comments has been addressed. Congratulations to the authors for the great work.

Reviewer #3 (Remarks to the Author):

I'd like to thank the authors for making the effort to address my concerns. Most of my concerns have been addressed. I have two minor points:

1. The authors have added Seurat results on Stereo-seq data in Supplementary figure 2a with different resolution parameters. But it is not mentioned or discussed in the manuscript what happens when changing the resolution.
2. Apart from Supplementary figure 2a, some other supplementary figures are not cited in the manuscript.

Response to Reviewer #1

The authors went through great length to address all the points I raised, and I think, also based on the other reviewers' comments, the manuscript has much improved. As such, I believe the manuscript can now be accepted as is.

Response: We are happy the reviewer agrees that the concerns have been successfully addressed, and are grateful for their help in significantly improving the manuscript.

Response to Reviewer #2

All my comments have been addressed. Congratulations to the authors for the great work.

Response: We appreciate the reviewer's kind words for our manuscript, as well as their efforts and insightful comments during the review process.

Response to Reviewer #3

I'd like to thank the authors for making the effort to address my concerns. Most of my concerns have been addressed. I have two minor points:

1. The authors have added Seurat results on Stereo-seq data in Supplementary figure 2a with different resolution parameters. But it is not mentioned or discussed in the manuscript what happens when changing the resolution.

2. Apart from Supplementary figure 2a, some other supplementary figures are not cited in the manuscript.

Response: We thank the reviewer for these comments. We have added text to note that the observations do not change when altering the resolution parameter at Line 261-

264 on Page 11. We have added descriptions and citations in the manuscript for all Supplementary Figures in the main text highlighted in red at Line 169-170 Page 7, Line 227-229 Page 9, Line 345-347 Page 14, Line 592-594 Page 24.